# Early detection of esophageal cancer: Evaluating AI algorithms with multi-institutional narrowband and white-light imaging data

**Young Seo Baik[1]☯, Hannah Lee[2]☯, Young Jae Kim[3], Jun-Won Chung[2]\*, Kwang Gi Kim[4]\***

**1** Department of Biomedical Engineering, Gachon University, Seongnam-si, Gyeonggi-do, Republic of Korea, **2** Division of Gastroenterology, Department of Internal Medicine, Gachon University Gil Medical Center, Incheon, Republic of Korea, **3** Department of Gachon Biomedical & Convergence Institute, Gachon University Gil Medical Center, Incheon, Republic of Korea, **4** Department of Biomedical Engineering, College of IT Convergence, Gachon University, Seongnam-si, Gyeonggi-do, Republic of Korea

☯ These authors contributed equally to this work.
\* kimkg@gachon.ac.kr (KGK); junwonchung@daum.net (J-WC)

## Abstract

Esophageal cancer is one of the most common cancers worldwide, especially esophageal squamous cell carcinoma, which is often diagnosed at a late stage and has a poor prognosis. This study aimed to develop an algorithm to detect tumors in esophageal endoscopy images using innovative artificial intelligence (AI) techniques for early diagnosis and detection of esophageal cancer. We used white light and narrowband imaging data collected from Gachon University Gil Hospital, and applied YOLOv5 and RetinaNet detection models to detect lesions. The models demonstrated high performance, with RetinaNet achieving a precision of 98.4% and sensitivity of 91.3% in the NBI dataset, and YOLOv5 attaining a precision of 93.7% and sensitivity of 89.9% in the WLI dataset. The generalizability of these models was further validated using external data from multiple institutions. This study demonstrates an effective method for detecting esophageal tumors through AI-based esophageal endoscopic image analysis. These efforts are expected to significantly reduce misdiagnosis rates, enhance the effective diagnosis and treatment of esophageal cancer, and promote the standardization of medical services.

## Introduction

Esophageal cancer is the eighth most common cancer worldwide and ranks among the top 10 fatal cancers [1]. However, almost all patients with esophageal adenocarcinoma are diagnosed at the end of the disease, and their prognosis is poor [2]. Currently, most diagnoses of esophageal squamous epithelial cell carcinoma are made using white-light imaging (WLI) endoscopy, and if dysplastic tissue is detected early, it can be treated with endoscopic mucosal resection and radio-frequency ablation [3]. Therefore, early detection and diagnosis are important for the survival and prognosis of patients with esophageal cancer [4]. Early diagnosis using WLI alone is difficult [5]. Instead of the iodine staining method, which induces

**Data availability statement:** The training data, specifically the image data, cannot be shared publicly due to the nature of medical data and as this study was conducted with the data that include sensitive personal information. Therefore, we are unable to open the dataset that was used for training with the imposing of the Institutional Review Board. Data used for training are available from the ethics committee (contact via email: irb@gilhospital.com) for researchers who meet the criteria for access to confidential data.

**Funding:** This work was supported by a grant from the Korean Gastrointestinal Endoscopy Research Foundation (2021 Investigation Grant), and by the Gachon University Gil Medical Center (Grant number: FRD2022-12), and by the Gachon University research fund of 2023(GCU-202308020001).

**Competing interests:** The authors have declared that no competing interests exist.

problems such as chest pain, discomfort, and increased procedure time, a useful technique was used to identify the structure by emphasizing the microvessels on the surface of the esophageal squamous cell carcinoma using narrowband imaging (NBI) [6]. NBI can help detect and diagnose early esophageal squamous epithelial cell carcinoma [7]. The complexity of esophageal squamous epithelial cell carcinoma, characterized by its much more refined and intricate shape compared to polypoid lesions, makes its accurate detection all the more challenging [8]. The use of conventional endoscopes is limited because they cannot easily discern the subtle changes in the initial lesion nor the number of biopsies, and lack high-definition imaging [9]. In addition, variability arises because of repeated and varied factors (experience, condition, fatigue, and mistakes) for the most important lesions [10]. Hence, the diagnostic accuracy of endoscopy may be reduced, and variations may occur depending on the diagnosis made by an inexperienced specialist [11,12]. Thus, diagnostic assistance using artificial intelligence (AI) technology is needed to improve the quality of healthcare services and reduce the occurrence of medical errors by diagnosing and assisting medical staff in the medical field [13].

Recently, AI technology utilizing deep learning (DL) with Convolutional Neural Networks (CNNs) has been applied in the medical field for the detection of various lesions in endoscopic images [14]. It has shown excellent results in the diagnosis and detection of lesions in the stomach, small intestine, and colon. Diagnosis using AI can help medical staff detect lesions early [15–17]. Wang et al. developed a deep learning algorithm to evaluate the difference in polyp and adenoma detection performance through colonoscopy and validated its effectiveness, achieving a sensitivity of 94.3% and specificity of 95.9% [18]. Similarly, Xu et al. designed an architecture for real-time classification and detection of gastric polyps through gastroscopy, achieving 100% sensitivity and 95.4% specificity, with excellent performance in detecting small polyps [19]. For oesophageal cancer, Goda et al. showed that magnified endoscopy with narrow-band imaging had a sensitivity of 78% and specificity of 95%, comparable to non-magnified high-resolution endoscopy (sensitivity 72%, specificity 92%) and high-frequency endoscopic ultrasound (sensitivity 83%, specificity 89%), and predicted the depth of invasion of superficial oesophageal squamous cell carcinoma, reducing the risk of overestimation by 25% compared to other techniques [20]. Nakagawa et al. found that an AI system using a single-shot multi-detector architecture to assess superficial squamous cell carcinoma achieved a sensitivity of 90.1%, specificity of 95.8%, and accuracy of 91%, which was similar to that of an experienced endoscopist, who achieved a sensitivity of 89.8, specificity of 88.3%, and accuracy of 89.6% [21]. Although there have been many CNN-based studies on lesion detection and diagnosis in various organs, medical data for esophageal squamous cell carcinoma is still limited compared to other datasets, which has led to problems such as overfitting and poor performance on new lesion images [22]. Wang et al. reported that Linked Colour Imaging had a specificity of 92.4% and sensitivity of 83.7% for oesophageal squamous cell carcinoma screening, which was similar to Lugol Chromoendoscopy with a specificity of 87% and sensitivity of 90.7%, and was promising for screening for squamous cell carcinoma and precancerous lesions in the general population with a much shorter procedure time [23]. However, due to these technical difficulties, few studies have compared and evaluated white-light and narrowband images by applying deep learning for esophageal squamous epithelial cell carcinoma. Therefore, the usefulness of applying detection methods needs to be evaluated and analyzed by collecting various esophageal squamous epithelial cell cancer data on white-light and narrowband images of AI-based multicenters.

In this study, we propose an AI algorithm that assists medical staff in the early detection and diagnosis of esophageal squamous epithelial cell carcinoma by analyzing esophageal

endoscopy information based on collected data. In addition, to evaluate generalizability, the performance of the system was verified using multicenter data [24]. This system can be a useful tool for warning medical staff when dysplastic lesions are detected during esophageal endoscopy by overcoming the lack of generalizability compared with the results detected by endoscopy in one institution [25]. A deep learning algorithm for early detection of esophageal cancer, utilizing multicenter data from narrowband and white-light imaging, can be an excellent method with the potential to enhance the efficiency and accuracy of diagnosis and treatment.

## Materials and methods

### Data acquisition and preprocessing

The data used in this study were obtained from 2,674 still images of 619 patients who underwent esophageal endoscopy (WLI) from January 2016 to June 2020 at Gachon University Gil Hospital and 480 still images of 121 patients who underwent esophageal endoscopy (NBI). This study received approval from the Gachon University Gil Hospital Clinical Research Ethics Review Committee, and the need for informed consent was waived due to the retrospective nature of the study (IRB No. GDIRB2020-316). The data access date for research purposes began on January 15, 2023, and continued until the end of the study. All experimental protocols were performed in accordance with the relevant guidelines and regulations of the Declaration of Helsinki.

To prevent patient-level data leakage, all images from a single patient were assigned exclusively to one of the training, validation, or test sets. Patient IDs were used to group images, ensuring no overlap between datasets. This approach guarantees that the model's performance metrics reflect its ability to generalize to unseen patient data. In the case of the WLI learning dataset, 2,674 sheets of normal, tumor-free, and tumor data from 619 patients were analyzed in a ratio of 8:1:1, divided into 1,925 sheets of learning data from 477 patients, 347 sheets of verification data from 60 patients, and 402 sheets of evaluation data from 82 patients, as shown in Table 1. In the case of the NBI learning dataset, based on the collected data, the dataset consisted of 480 sheets of 121 patients with tumors, divided into 374 sheets of learning data from 97 patients, 37 sheets of verification data from 12 patients, and 69 sheets of evaluation data from 12 patients, as shown in Table 1. As the WLI and NBI data had different horizontal and vertical ratios, all the images were resized to $640 \times 640$ pixels and used in the experiment.

To learn and verify the deep learning model, the ground truth was obtained by labeling the location of the lesion. In this study, the regions of interest (ROIs) were annotated by a gastroenterologist with more than 10 years of clinical experience to ensure accuracy and reliability. Using the ImageJ labeling software, a region of interest in the form of a rectangle, including the entire shape of the tumor, was displayed through a specialized inspection process. Among the collected data, one image was randomly selected for each image type and is presented in Fig 1 with labels.

**Table 1. Number of images collected for WLI/NBI data.**

|  | White-light imaging | | | Narrowband imaging | |
|---|---|---|---|---|---|
|  | **Normal** | **Cancer** | **Total** | **Cancer** | **Total** |
| **Train** | 238 (1,037) | 239 (888) | 477 (1,925) | 97 (374) | 97 (374) |
| **Validation** | 30 (178) | 30 (169) | 60 (347) | 12 (37) | 12 (37) |
| **Test** | 53 (201) | 29 (201) | 82 (402) | 12 (69) | 12 (69) |

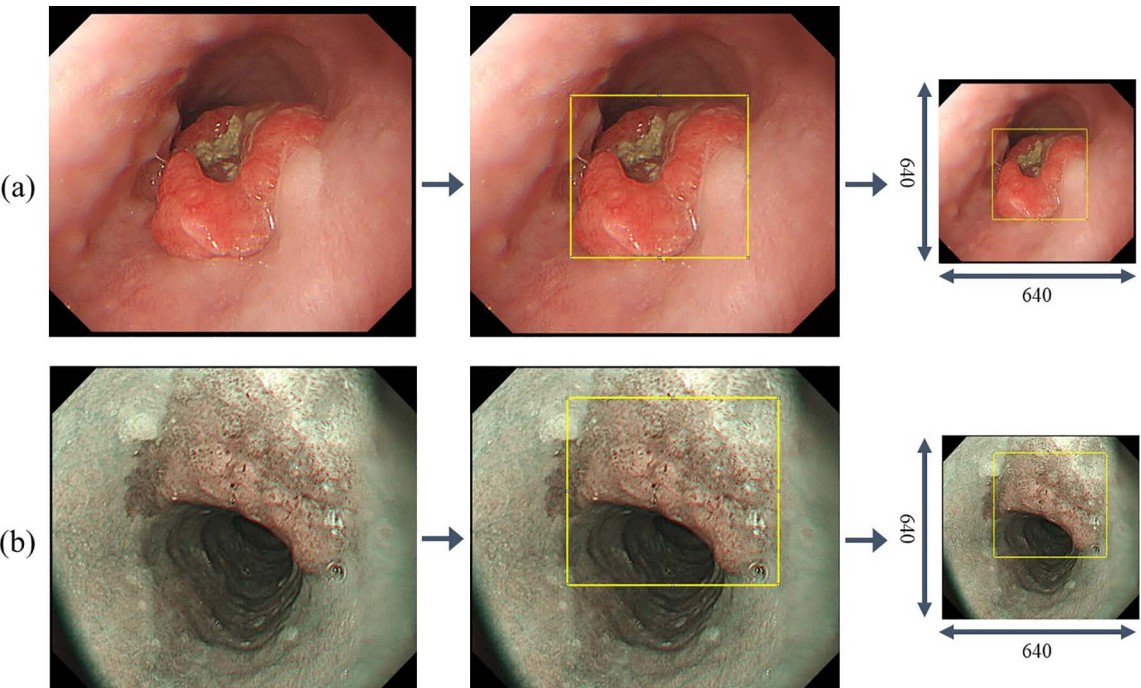

**Fig 1. Labeling data for regions of interest.** (a) white-light imaging (WLI). (b) narrow-band imaging (NBI).

## Configuration of deep learning detection models

The YOLOv5 model, which is a single-stage object detection framework, was applied to detect tumors in esophageal endoscopy images, as shown in Fig 2 [26]. YOLO, which was used as a feature extraction model, predicts multiple areas in an image simultaneously using one convolution network and analyzes the class probability through a single regression. Its learning speed is faster because there is no complex pipeline in the model, and its detection performance is better than that of the R-CNN series model [27]. The parameters were normalized as the width and height of the image based on the ratio of the width and height of the bounding boxes. The prediction result of YOLO determines the final prediction label based on the prediction annotation coordinates and class probability.

Second, the RetinaNet model of the single-stage object detection framework, which introduced the concept of focus loss for the first time, was applied, as shown in Fig 2 [28]. RetinaNet comprises a backbone network and two subnetworks that perform classification and bounding-box regression, respectively. The backbone, which is a publicly available open convolution network, calculates a convolution feature map for the entire area of the input image. The first subnet performed object classification using the convolution output of the backbone. The second subnet obtained the coordinates of the bounding box (offset between the anchor and the reference point) through convolution at the backbone output.

## Experiment setup

The experimental environment of this study used a system consisting of two NVIDIA GeForce GTX 2080 Ti (NVIDIA, Santa Clara, CA, USA) graphics processing units and an Intel® Xeon® Gold 6238 CPU @ 2.10 GHz and 32 GB of RAM and was executed on the Ubuntu 16.04 operating system. TensorFlow (version 1.14.0), PyTorch (version 1.7.1), Keras (version 2.2.4), and Python (version 3.6.12) were used for the deep learning.

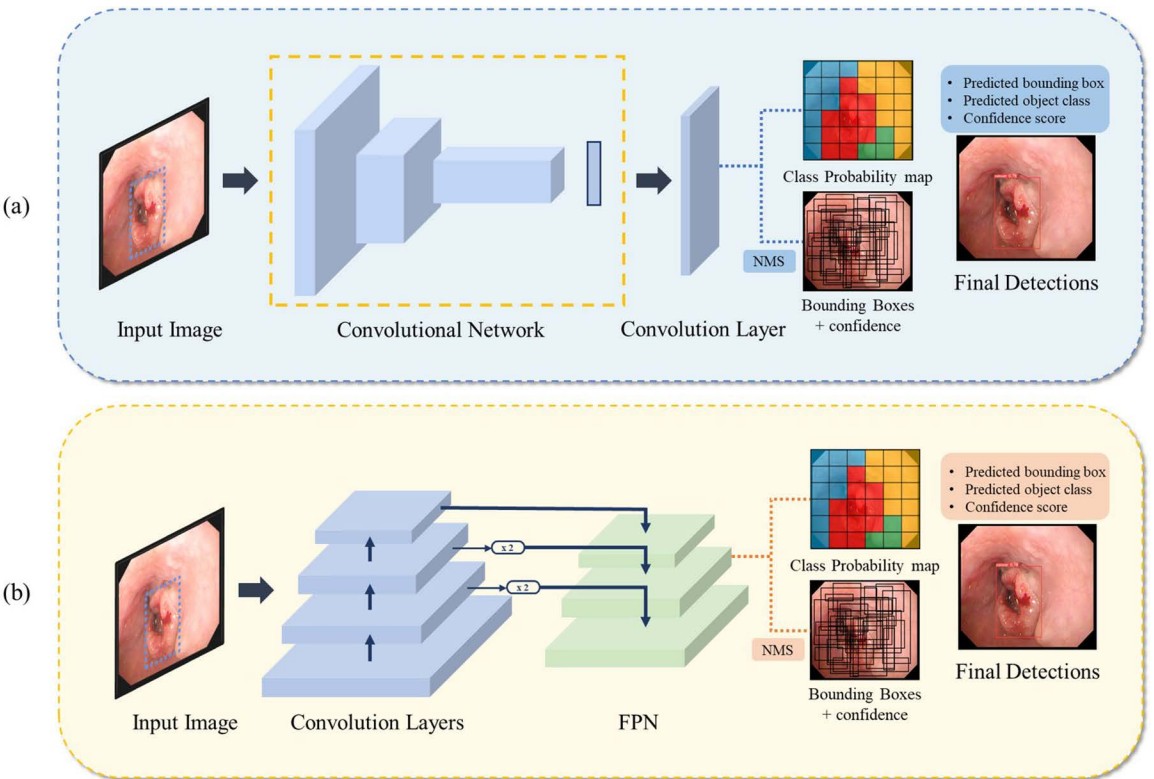

**Fig 2. Architecture for Tumor detection in esophageal Endoscopy images.** (a) YOLOv5. (b) RetinaNet.

## Deep learning model parameters and evaluation index

In this study, YOLOv5l was used among five model sizes, from YOLOv5sl to YOLOv5xl. The prediction and learning conditions of YOLOv5 were set to 200 epochs and a batch size of 16 using an image size optimization algorithm of 640×640 and a learning rate of 1e-3 (Adam). An early stopping algorithm was applied to prevent overfitting. To compute the final loss in YOLOv5, we used the ComputeLoss function, which integrates class loss, objectivity loss, and bounding box loss. The prediction and learning conditions of RetinaNet were set to 200 epochs and a batch size of 1 using an image size (learning rate) optimization algorithm of 640×640 and a learning rate of 1e-5. To address class imbalance in RetinaNet, we employed Focal Loss. An early stopping algorithm was applied to prevent overfitting. Furthermore, we utilized the ReLU activation function in the backbone layer and the Sigmoid activation function in the final layer for classification.

The learned detection model was compared and analyzed using performance evaluation indicators such as precision, sensitivity, and false positives per image (FPPI). The confidence score is an index that can determine the class classification and position detection results of the detected boundary box and is obtained by multiplying the probability of the class predicted by the model to be correct for the object detected with the intersection over union (IoU) value. A true positive (TP) is the case of correctly detecting the tumor location obtained through the tumor location detection model; a false positive (FP) is the case of detecting the location without the tumor; and a false negative (FN) is the case of failure to detect the tumor. Using the confusion matrices, several performance indicators were calculated as (1), (2), (3), and (4). Precision is the ratio of what the model correctly identifies

among the predictions of a lesion, and sensitivity is the ratio of what the model predicts among the data with actual lesions. The ratio of the number of FP images detected per image was used for the FPPI, which indicates that the scale fluctuation was very wide, depending on the data. To compare and evaluate the detection performance, the change according to the adjustment of the parameter through the precision–recall curve with an exchange relationship was graphically represented.

$$Precision = \frac{TP}{TP + FP} \tag{1}$$

$$Sensitivity = \frac{TP}{TP + FN} \tag{2}$$

$$FPPI = \frac{FP}{Number\ of\ images} \tag{3}$$

$$IoU = \frac{Avea\ of\ Overlap}{Area\ of\ Union} \tag{4}$$

## Additional experiments

To test the generalizability of the model, external validation was performed using images acquired from patients who underwent esophageal endoscopy (WLI and NBI) at Kyunghee Medical Center, Korea University Anam Hospital, and Hallym University Sacred Heart Hospital. For the WLI dataset, the detection performance of the model was tested using data from 112 tumors from Kyunghee Medical Center, 353 tumors from Korea University Anam Hospital, and 23 tumors from Hallym University Sacred Heart Hospital. For the NBI learning dataset, the detection performance of the model was tested using data from 29 tumors from Kyunghee University Hospital, 192 tumors from Korea University Anam Hospital, and 13 tumors from Hallym University Sacred Heart Hospital.

## Results and discussion

### Performance comparison of detection network models

Based on the presence or absence of lesions in the esophageal endoscopy image, two classes of data—normal and with lesions—were designated, and the results detected by the AI model were analyzed. Esophageal lesions were defined as true, and no esophageal lesions were defined as false. When the IoU value between the prediction and correct answer areas was 0.5, the prediction was considered successful.

To confirm the precision, sensitivity, and FPPI according to the compliance threshold, the performances of the esophageal cancer detection models YOLOv5 and RetinaNet were compared and analyzed for the detection results with a threshold value of 0.1 or more, as shown in Table 2. In the WLI dataset, the YOLOv5 model detected images with a precision of 93.7%, a sensitivity of 89.9%, and an FPPI of 6%. The RetinaNet model detected images with a precision of 96.1%, a sensitivity of 88.4%, and an FPPI of 3.5%. In the NBI dataset, the YOLOv5 model detected images with a precision of 86.5%, a sensitivity of 84.0%, and an FPPI of 13%. The RetinaNet model detected images with a precision of 98.4%, a sensitivity of 91.3%, and an FPPI of 1.4%. From the WLI dataset, 402 evaluation data points were obtained,

composed of 201 normal data points without tumors and 201 data points with tumors, and the performance of the detection model was evaluated. In the YOLOv5 model, 179 of the 201 data points with tumors were determined as data with tumors (TP), and 20 were determined as data without tumors (FN). Moreover, 12 of the 201 normal data points without tumors were determined to have tumors (FP). In the RetinaNet model, 176 of the 201 data points with tumors were determined as data with tumors (TP), and 23 were determined as data without tumors (FN). Moreover, 7 of the 201 normal data points without tumors were determined as data with tumors (FP). By showing an example of image detection in Fig 3, the true detection results of the tumor location predicted by the detection model and the actual tumor location can be confirmed. In the NBI dataset, 69 evaluation data points with tumors were constructed to evaluate the performance of the detection model. In the YOLOv5 model, 58 of the 69 data points with tumors were determined as data with tumors (TP), and 11 were determined as data without tumors (FN). Nine normal data points without tumors were identified as data points with tumors (FP). In the RetinaNet model, 63 of the 69 data points with tumors were

**Table 2. Performance evaluation metrics for detection models based on confidence thresholds from internal data.**

| | Model | TP | FN | FP | Precision (95% CI) | Sensitivity (95% CI) | FPPI (95% CI) |
|---|---|---|---|---|---|---|---|
| **White-light imaging** | YOLOv5 | 179 | 20 | 12 | 0.937 (0.892–0.951) | 0.899 (0.857–0.924) | 0.06 (0.043–0.075) |
| | RetinaNet | 176 | 23 | 7 | 0.961 (0.88–0.984) | 0.884 (0.80–0.954) | 0.035 (0.016–0.043) |
| **Narrowband imaging** | YOLOv5 | 58 | 11 | 9 | 0.865 (0.824–0.913) | 0.840 (0.763–0.88) | 0.13 (0.07–0.267) |
| | RetinaNet | 63 | 6 | 1 | 0.984 (0.951–0.99) | 0.913 (0.842–0.944) | 0.014 (0.008–0.035) |

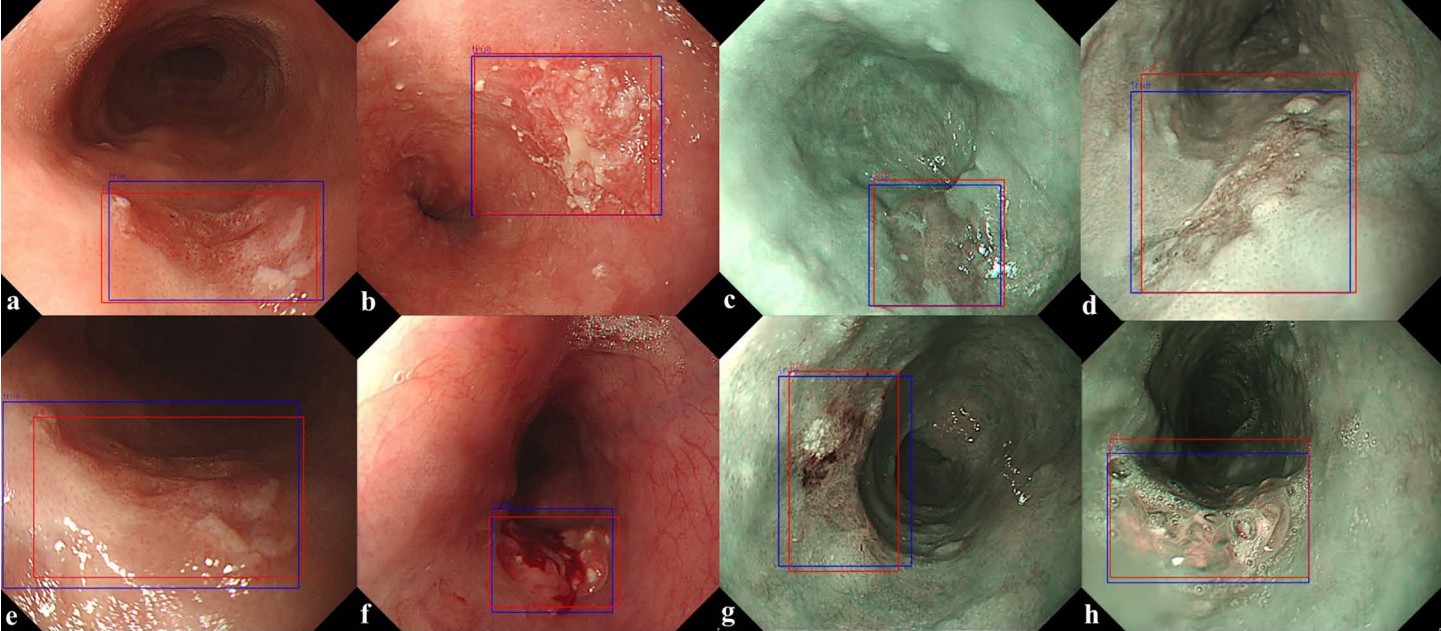

**Fig 3. TP predictions from a trained model for tumor location detection.** (a–d) YOLOv5, (e–h) RetinaNet. (blue color: ground truth, red color: predicted result).

determined as data with tumors (TP), and six were determined as data without tumors (FN). A normal datum without tumors was determined to be a datum with tumors (FP). As shown in Fig 4, the detection of FP and FN results for the tumor location predicted by the detection model and the actual tumor location can be confirmed from the internal data. FP results were obtained because of the prediction of shadows from normal data as lesions, which accounted for almost all cases. In addition, as shown in Fig 4b, when the lesion was very small and far away, the nearby crystal area was predicted to be an FP. The main cause of the FN predicted results was esophageal inflammation in the mucous membrane, as shown in Fig 4c. In addition, even when the lesion occupied the entire area, as shown in Fig 4d, it could not be predicted. Fig 5 shows the overall performance of the model with a precision–recall curve for

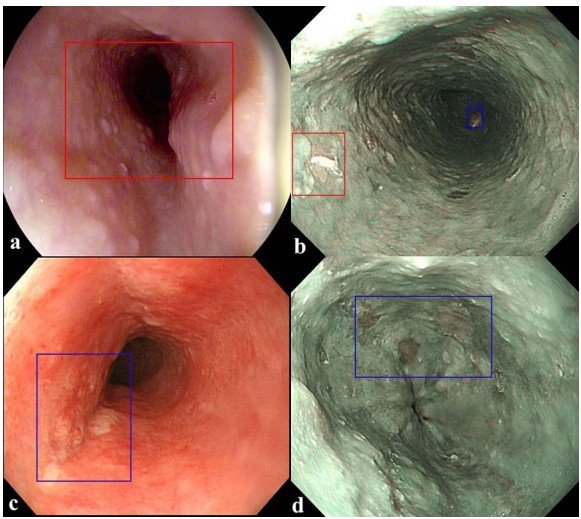

**Fig 4. Prediction results of a trained model for detecting the location of a tumor.** (a, b) False positive, FP. (c, d) False negative, FN. (blue color: ground truth, red color: predicted result).

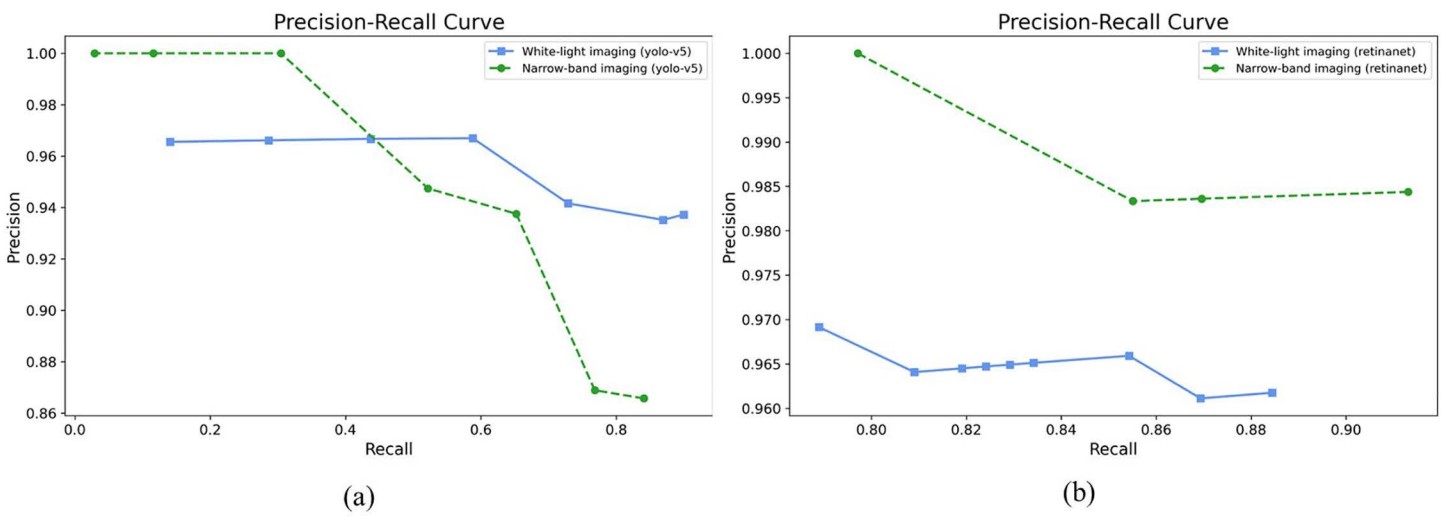

**Fig 5. Precision–recall curves obtained using the detection model for internal data.** (a) YOLOv5. (b) RetinaNet.

the internal data. In general, the closer the curve is to the upper-right corner, the better the performance of the model. The two detection models identify the positive classes well and simultaneously consider the number of negative classes incorrectly classified as positive. In the detection model, recall with a low FN ratio was more important.

Through external verification, according to the presence or absence of lesions in the esophageal endoscopic image, two classes of data were designated as normal without lesions and with lesions, and the results detected by the AI model were analyzed. To confirm the precision, sensitivity, and FPPI according to the compliance threshold, the performances of the esophageal cancer detection models YOLOv5 and RetinaNet were compared and analyzed for the detection results with a threshold value of 0.1 or more, as shown in Table 3. In the WLI dataset, the YOLOv5 model detected images with a precision of 83.4%, a sensitivity of 79.4%, and an FPPI of 15.8%. The RetinaNet model detected images with a precision of 88.3%, a sensitivity of 70.2%, and an FPPI of 9.2%. In the NBI dataset, the YOLOv5 model detected images with a precision of 85.6%, a sensitivity of 71.3%, and an FPPI of 11.9%. The RetinaNet model detected images with a precision of 88.3%, a sensitivity of 81.1%, and an FPPI of 10.6%. In the WLI dataset, 488 evaluation data points with tumors were constructed to evaluate the performance of the detection model. In the YOLOv5 model, 387 of the 488 tumors were identified as data with tumors (TP) and 100 as data without tumors (FN). Moreover, 77 normal data points without tumors were identified as data with tumors (FP). In the RetinaNet model, 342 of the 488 tumors were determined as data with tumors (TP), and 145 were determined as data without tumors (FN). Moreover, 45 normal data points without tumors were identified as data with tumors (FP). By showing an example of image detection in Fig 6, the true detection results of the tumor location predicted by the detection model and the actual tumor location can be confirmed.

In the NBI dataset, 288 evaluation data points with tumors were constructed to evaluate the performance of the detection model. In the YOLOv5 model, 167 of the 288 tumors were identified as data with tumors (TP), and 67 were determined as data without tumors (FN). Moreover, 28 normal data points without tumors were identified as data points with tumors (FP). In the RetinaNet model, 190 of the 288 tumors were determined as data with tumors (TP), and 44 were determined as data without tumors (FN). Moreover, 25 normal data points without tumors were identified as data points with tumors (FP). As shown in Fig 7, the detection of FP and FN results for the tumor location predicted by the detection model and the actual tumor location can be confirmed from external data. FP results were obtained because of the prediction of shadows from normal data as lesions, which accounted for almost all cases. In addition, as shown in Fig 7b, the overall lesion was predicted as an FP. The main cause of the result predicted as an FN was the presence of only a part of the lesion, as shown in Fig 7c. The case of esophageal inflammation of the mucous membrane shown in Fig 7d could not be

**Table 3. Performance evaluation metrics for detection models based on confidence thresholds from external data.**

|  | Model | TP | FN | FP | Precision (95% CI) | Sensitivity (95% CI) | FPPI (95% CI) |
|---|---|---|---|---|---|---|---|
| **White-light imaging** | YOLOv5 | 387 | 100 | 77 | 0.834 (0.761–0.88) | 0.794 (0.756–0.85) | 0.158 (0.086–0.194) |
|  | RetinaNet | 342 | 145 | 45 | 0.883 (0.82–0.956) | 0.702 (0.657–0.782) | 0.092 (0.057–0.18) |
| **Narrowband imaging** | YOLOv5 | 167 | 67 | 28 | 0.856 (0.817–0.893) | 0.713 (0.68–0.76) | 0.119 (0.007–0.142) |
|  | RetinaNet | 190 | 44 | 25 | 0.883 (0.853–0.947) | 0.811 (0.766–0.89) | 0.106 (0.08–0.181) |

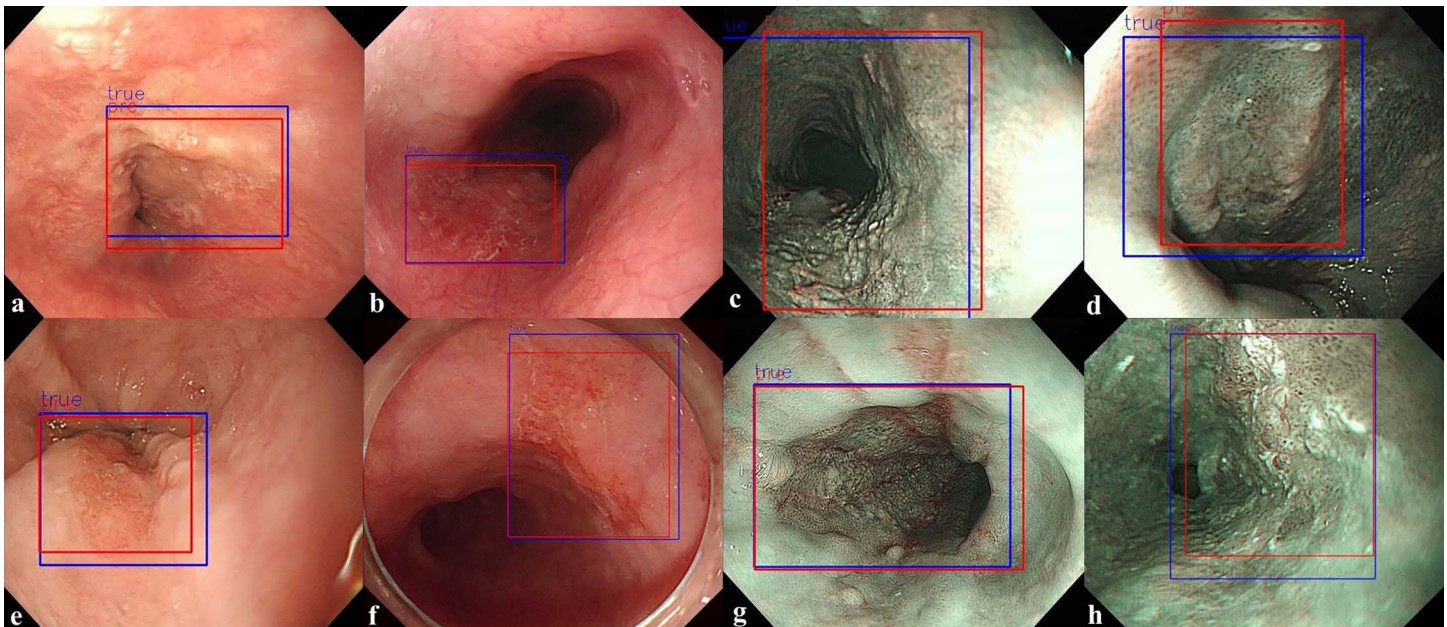

**Fig 6. TP predictions from a trained model for tumor location detection.** (a–d) YOLOv5, (e–h) RetinaNet. (blue color: ground truth, red color: predicted result).

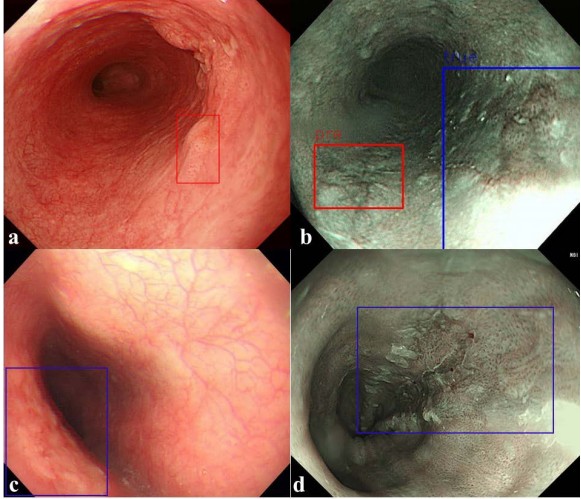

**Fig 7. Prediction results of a trained model for detecting the location of a tumor.** (a, b) False positive, FP. (c, d) False negative, FN. (blue color: ground truth, red color: predicted result).

predicted to be a lesion. Fig 8 shows the overall performance of the model with a precision–recall curve for external data. In general, the closer the curve is to the upper-right corner, the better the performance of the model. The two detection models identify the positive classes well and simultaneously consider the number of negative classes incorrectly classified as positive. In the detection model, recall with a low FN rate is more important. High precision can lead to low recall, indicating that the model misses most of the tumor data.

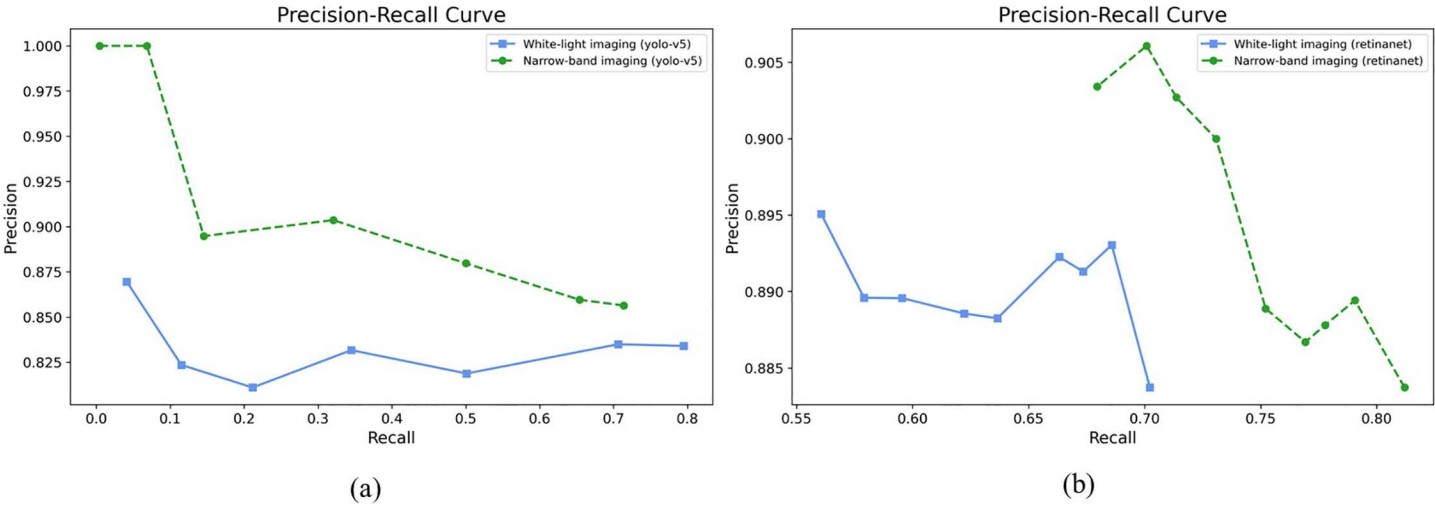

**Fig 8. Precision–recall curves obtained using the detection model for external data.** (a) YOLOv5. (b) RetinaNet.

## Conclusion

In this study, an AI algorithm was proposed to detect the location of esophageal tumors using multicenter data from esophageal endoscopic WLI and NBI tests. Normal data without areas of interest were also learned so that the AI learning model could reduce FPs and increase sensitivity in normal data. To increase sensitivity, which is particularly important in detection research, it is necessary to find as many objects as possible in the data. Some images have no special elements; hence, the FPPI is 0. However, other images have several objects that can be recognized as objects; hence, the FPPI can have a large value. The study was conducted by carefully analyzing the performance indicators. The confidence threshold was set to 0.25, and the IoU threshold was set to 0.5, to prevent the removal of additional bounding boxes. The detection model showed high precision and sensitivity, and normal and tumor data were classified with high accuracy. In addition, data were collected from various institutions to verify the relatively high generalization performance. The established database can be used as important data for CAD research and algorithm development for future endoscopies. Most of the related cases mentioned were mainly polyps to detect and diagnose lesions in various organs; however, in this study, lesion detection was performed in white-light and narrow-band images by analyzing not only polyps but also superficial esophageal cancer [14–21].

This revised section highlights the advantages of the proposed method, including its strong generalization capabilities across multicenter datasets, its high precision and sensitivity for both WLI and NBI data, and its significant contribution to the early diagnosis of esophageal cancer. However, the limitations of the study are also addressed, particularly the potential decline in model performance when encountering unseen datasets with novel artifacts or rare tumor types. These challenges are acknowledged, and potential strategies for overcoming them are discussed.

To improve the performance of the model in the future, relearning through additional data collection and cross-verification processes is required to improve its reliability. In addition, the performance needs to be optimized by fine-tuning the parameters of the AI algorithm based on feedback from the verification [29]. It is determined that the area of the tumor location will stand out owing to the mitigation or overcoming of the limitations of the existing pre-processing, which will be helpful in learning performance. To detect morphological

tumors, the performance of the tumor detection model can be improved by increasing its clinical suitability by performing post-treatment separately [30].

Future research directions include extending the proposed approach to applications such as dermatological disease detection and abdominal organ segmentation. These extensions aim to demonstrate the versatility and effectiveness of the method in addressing challenges in other medical imaging domains. As a future work, the performance of the proposed method can be tested for the classification of dermatological diseases from dermoscopy images because detection of skin lesions is challenging and an AI-based effective method is still desired in this field despite some recent approaches [31–33]. Also, as another future work, the proposed method can be modified to achieve abdominal organ segmentation, such as the liver and kidneys, from grayscale medical images because noise and low contrasts make their segmentations difficult, and atlas or level set-based methods [34–39] are not always effective.

Incorporating synthetic algorithm technology, such as generative adversarial networks, to generate data from narrowband images could further improve the model by enabling predictions of lesion invasion depth. These efforts are expected to contribute to the effective diagnosis and treatment of esophageal cancer and promote the standardization of medical services.

This study demonstrated the feasibility of an effective method for detecting esophageal tumors in AI-based esophageal endoscopy images. By applying division and processing in frames based on real-time videos, the proposed method can be advantageously utilized in the current endoscopy environment. Algorithmic weight reduction and optimization technologies can also be implemented to enable real-time processing by improving the processing speed of each algorithm. These findings are expected to enhance the quality of medical services, enable more precise and rapid diagnosis, and reduce the misdiagnosis rate by providing robust diagnostic support to medical staff.

## Author contributions

**Conceptualization:** Young Jae Kim.

**Data curation:** Hannah Lee, Jun-Won Chung.

**Formal analysis:** Young Seo Baik, Jun-Won Chung.

**Funding acquisition:** Young Seo Baik, Jun-Won Chung.

**Investigation:** Hannah Lee.

**Methodology:** Young Seo Baik, Young Jae Kim.

**Supervision:** Young Jae Kim, Jun-Won Chung, Kwang Gi Kim.

**Validation:** Young Seo Baik.

**Visualization:** Young Seo Baik.

**Writing – original draft:** Young Seo Baik.

**Writing – review & editing:** Young Seo Baik, Young Jae Kim, Jun-Won Chung, Kwang Gi Kim.

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
