## [Decision Letter · Decision Letter 0]

4 Dec 2024

PONE-D-24-42374Early detection of esophageal cancer: evaluating ai algorithms with multi-institutional narrorwband and white-light imaging dataPLOS ONE

Dear Dr. Kim,

Thank you for submitting your manuscript to PLOS ONE. After careful consideration, we feel that it has merit but does not fully meet PLOS ONE’s publication criteria as it currently stands. Therefore, we invite you to submit a revised version of the manuscript that addresses the points raised during the review process.

We look forward to receiving your revised manuscript.

Kind regards,

Hirenkumar Kantilal Mewada

Academic Editor

PLOS ONE

Journal Requirements:

[This work was supported by a grant from the Korean Gastrointestinal Endoscopy Research Foundation (2021 Investigation Grant), and by the Gachon University Gil Medical Center (Grant number: FRD2022-12), and by the Gachon University research fund of 2023(GCU-202308020001)].

Additional Editor Comments:

One or more of the reviewers has recommended that you cite specific previously published works. Members of the editorial team have determined that the works referenced are not directly related to the submitted manuscript. As such, please note that it is not necessary or expected to cite the works requested by the reviewer.

Reviewers' comments:

Reviewer's Responses to Questions

**Comments to the Author**

1. Is the manuscript technically sound, and do the data support the conclusions?

Reviewer #1: Yes

Reviewer #2: Partly

Reviewer #3: Partly

2. Has the statistical analysis been performed appropriately and rigorously? 

Reviewer #1: Yes

Reviewer #2: Yes

Reviewer #3: No

3. Have the authors made all data underlying the findings in their manuscript fully available?

Reviewer #1: Yes

Reviewer #2: No

Reviewer #3: Yes

4. Is the manuscript presented in an intelligible fashion and written in standard English?

Reviewer #1: Yes

Reviewer #2: Yes

Reviewer #3: Yes

5. Review Comments to the Author

Reviewer #1: In this paper, the authors propose a method to detect tumors in esophageal endoscopy images using innovative artificial intelligence techniques for early diagnosis and detection of esophageal cancer.

Esophageal cancer is one of the most common cancers worldwide, especially esophageal squamous cell carcinoma, which is often diagnosed at a late stage and has a poor prognosis.

Therefore, the topic handled in this work is important.

The proposed approach can be helpful for other researchers. However, the following revisions are required to improve the quality of the paper;

1) In the abstract section, some important numerical results should be added to inform the readers about the performance and effectiveness of the proposed method.

2) Grammatical mistakes should be corrected.

3) The loss and activation functions used in this work should be written.

4) The last section should be "Conclusion". In the discussion section, the advantages/superiority of the proposed approach should be explained.

Also, limitations/drawbacks should be added. Possible extensions of this work and future works should be added.

In future work, the effectiveness of the proposed approach can be investigated for lesion detection from dermoscopy images, which are colored images, and an effective AI-based method is needed in this field although there are recent advancements.

Also, as another future work, the proposed method can be modified to achieve abdominal organ segmentation, such as the liver and kidneys, from grayscale medical images because noise and low contrasts make their segmentations difficult, and atlas or level set-based methods are not always effective.

Therefore, the following statements should be added;

"As a future work, the performance of the proposed method can be tested for the classification of dermatological diseases from dermoscopy images because detection of skin lesions is challenging and an AI-based effective method is still desired in this field despite some recent approaches [R1-R3].

Also, as another future work, the proposed method can be modified to achieve abdominal organ segmentation, such as the liver and kidneys, from grayscale medical images because noise and low contrasts make their segmentations difficult, and atlas or level set-based methods [R4-R9] are not always effective.

R1: Automated Skin Cancer Detection: Where We Are and The Way to The Future, https://doi.org/10.1109/TSP52935.2021.9522605

R2: Convolutional neural network based desktop applications to classify dermatological diseases, https://doi.org/10.1109/IPAS50080.2020.9334956

R3: Comparative evaluations of cnn based networks for skin lesion classification, Int.Conf.onon Computer Graphics, Visualization, Computer Vision and Image Processing (CGVCVIP)

R4: A comparative performance evaluation of various approaches for liver segmentation from SPIR images, https://doi.org/10.3906/elk-1304-36

R5: A neural network based kidney segmentation from MR images, https://doi.org/10.1109/ICMLA.2015.229

R6: A method for liver segmentation in perfusion MR images using probabilistic atlases and viscous reconstruction, https://doi.org/10.1007/s10044-017-0666-z

R7: A comparative evaluation for liver segmentation from spir images and a novel level set method using signed pressure force function, Phd thesis,  Izmir Institute of Technology

R8: An automatic level set based liver segmentation from MRI data sets, https://doi.org/10.1109/IPTA.2012.6469551

R9: Automatic kidney segmentation using Gaussian mixture model on MRI sequences, https://doi.org/10.1007/978-3-642-21747-0_4

Reviewer #2: You have done a commendable work but I believe the comments below can improve the work;

Comment #1: Check the typographical error in the last sentence before the conclusion of the introduction. You wrote "om" instead of "on." Then, go through the entire work to rectify typographical errors.

Comment #2: The part of the introduction containing the literature is scanty and must be elaborated on. There is a lot of work on esophageal cancer using AI, and I think you need to reference or add more of this coupled with the evaluation metric values attained.

Comment #3: It would like you to show the confusion matrices that you used to calculate TP, TN, FP, FN, precision, Sensitiviy etc. in the tables and in the text.

Comment #4: If this dataset has been used by any AI algorithm before, a comparative analysis with the authors' results and the existing models' results will help readers and researchers appreciate your efforts.

Comment #5: It would have been good if the Accuracy evaluation metric was also used to assess the models.

Comment #6: You indicated you were proposing an AI algorithm, but I saw the use of existing models and did not see your novelty in this study. Therefore, I would like you to highlight your novelty.

Comment #7: If possible, let's have a link to the dataset you used in this study.

Reviewer #3: The paper presents commendable work. However, several aspects require further clarification and elaboration to improve the transparency and impact of the study.

• Contributions

1. Can the authors provide a clear summary of the key contributions of their work? Highlighting the novelty and practical implications will help better understand the overall significance of the study.

• Dataset

1. Annotation Process: How were the annotations verified? Were domain experts involved in the process?

2. Data Leakage: Patient-level leakage is a critical concern in medical imaging datasets. Please detail the measures taken to prevent such leakage. For instance, were all images from a single patient assigned exclusively to either the training, validation, or test sets?

3. Dataset Size: Given the relatively small size of the dataset, were data augmentation techniques employed to increase its size? If yes, please describe the augmentation strategies used.

4. Class Imbalance: How did the authors address class imbalance in the dataset?

• Model Selection

1. What motivated the selection of YOLOv5 and RetinaNet for this study? Were alternative models considered?

2. Please provide details of any modifications made to the baseline architectures of YOLOv5 and RetinaNet.

• Performance Evaluation

1. Comparison with State-of-the-Art: The results appear promising; however, the authors are encouraged to provide a direct comparison with state-of-the-art methods. While datasets may differ, comparisons with studies using esophageal datasets (e.g. leveraging WLI or NBI) would provide additional context. A summary table would be helpful for this purpose.

2. Confusion Matrix: To better visualize the model's performance, confusion matrices should be included, detailing metrics such as TP/FP/TN/FN. This will provide clearer insights into class-wise behavior.

Addressing these points will enhance the clarity, transparency, and overall impact of the manuscript.

Considering PLOS ONE’s specific guidelines on code sharing ‘https://journals.plos.org/plosone/s/materials-and-software-sharing#loc-sharing-code’ for submissions involving author-generated code, I would encourage the authors to share their code in a way that supports the findings in the manuscript.

6. PLOS authors have the option to publish the peer review history of their article (what does this mean? ). If published, this will include your full peer review and any attached files.

**Do you want your identity to be public for this peer review?** For information about this choice, including consent withdrawal, please see our Privacy Policy .

Reviewer #1: No

Reviewer #2: No

Reviewer #3: No

---

## [Author Response · Author response to Decision Letter 0]

12 Jan 2025

Response to Reviewer 1

We sincerely thank the reviewer for their constructive feedback, which has significantly improved the quality of our manuscript. Below, we address each comment in detail:

1. In the abstract section, some important numerical results should be added to inform the readers about the performance and effectiveness of the proposed method.

Response

As per the reviewer's comments, we have added some important numerical results in the Abstract section to inform the reader about the performance and effectiveness of the proposed method. We have revised the abstract to include key numerical results that demonstrate the effectiveness of the proposed method. Specifically, we state that RetinaNet achieved 98.4% precision and 91.3% sensitivity on the NBI dataset and YOLOv5 achieved 93.7% precision and 89.9% sensitivity on the WLI dataset. This content has also been added to the paper. Thank you for mentioning these important points to improve the paper.

Modified and supplemented contents - Page 2, line 7~14

2. Grammatical mistakes should be corrected.

Response

We have thoroughly reviewed the manuscript and addressed all grammatical issues. The language throughout the manuscript has been refined to enhance clarity and readability. For example, ambiguous sentences in the abstract and discussion sections have been restructured for better coherence. This content has also been added to the paper. Thank you for leaving a comment to improve the completeness of this paper.

3. The loss and activation functions used in this work should be written.

Response

In the revised manuscript, we described the loss and activation functions in more detail. Specifically, we used Focal Loss for the RetinaNet model to address class imbalance and ComputeLoss loss for YOLOv5 to compute the final loss by integrating class loss, objectivity loss, and bounding box loss. In addition, we used ReLU activation function in the backbone layer and Sigmoid activation function in the final layer for classification. This content has also been added to the paper. Thank you for mentioning important points to improve the completeness of this paper.

Modified and supplemented contents - Page 8, line 21~22, Page 9, line 1~2, line 3~4

4. The last section should be "Conclusion". In the discussion section, the advantages/superiority of the proposed approach should be explained.

Also, limitations/drawbacks should be added. Possible extensions of this work and future works should be added.

In future work, the effectiveness of the proposed approach can be investigated for lesion detection from dermoscopy images, which are colored images, and an effective AI-based method is needed in this field although there are recent advancements.

Response

Based on the reviewer’s comments, the final section of the manuscript has been renamed ‘Conclusions’ to better reflect its content, and the Discussion section has been expanded and integrated with the Results section under the title ‘Results and Discussion’. This revised section highlights the advantages of the proposed method, including its strong generalization capabilities across multicenter datasets, its high precision and sensitivity for both white-light imaging (WLI) and narrow-band imaging (NBI) data, and its significant contribution to the early diagnosis of esophageal cancer. However, the limitations of the study are also addressed, particularly the potential decline in model performance when encountering unseen datasets with novel artifacts or rare tumor types. These challenges are acknowledged, and potential strategies for overcoming them are discussed. Furthermore, the manuscript has been enhanced to outline future research directions, including extending the proposed approach to applications such as dermatological disease detection and abdominal organ segmentation. These extensions are expected to demonstrate the versatility and effectiveness of the method in addressing challenges in other medical imaging domains. This content has also been added to the paper. Thank you for improving the completeness of this paper by leaving a comment regarding the shortcomings of this paper.

Modified and supplemented contents – Page 11, line 1, Page 16, line 1, line 17~22, Page 17, 6~9, 16~24, Page 18, line 1~2

5. Also, as another future work, the proposed method can be modified to achieve abdominal organ segmentation, such as the liver and kidneys, from grayscale medical images because noise and low contrasts make their segmentations difficult, and atlas or level set-based methods are not always effective.

Therefore, the following statements should be added;

"As a future work, the performance of the proposed method can be tested for the classification of dermatological diseases from dermoscopy images because detection of skin lesions is challenging and an AI-based effective method is still desired in this field despite some recent approaches [R1-R3].

Also, as another future work, the proposed method can be modified to achieve abdominal organ segmentation, such as the liver and kidneys, from grayscale medical images because noise and low contrasts make their segmentations difficult, and atlas or level set-based methods [R4-R9] are not always effective.

R1: Automated Skin Cancer Detection: Where We Are and The Way to The Future, https://doi.org/10.1109/TSP52935.2021.9522605

R2: Convolutional neural network based desktop applications to classify dermatological diseases, https://doi.org/10.1109/IPAS50080.2020.9334956

R3: Comparative evaluations of cnn based networks for skin lesion classification, Int.Conf.onon Computer Graphics, Visualization, Computer Vision and Image Processing (CGVCVIP)

R4: A comparative performance evaluation of various approaches for liver segmentation from SPIR images, https://doi.org/10.3906/elk-1304-36

R5: A neural network based kidney segmentation from MR images, https://doi.org/10.1109/ICMLA.2015.229

R6: A method for liver segmentation in perfusion MR images using probabilistic atlases and viscous reconstruction, https://doi.org/10.1007/s10044-017-0666-z

R7: A comparative evaluation for liver segmentation from spir images and a novel level set method using signed pressure force function, Phd thesis,  Izmir Institute of Technology

R8: An automatic level set based liver segmentation from MRI data sets, https://doi.org/10.1109/IPTA.2012.6469551

R9: Automatic kidney segmentation using Gaussian mixture model on MRI sequences, https://doi.org/10.1007/978-3-642-21747-0_4

Response

Based on the reviewer's comments, we have incorporated the following suggested future research directions into the paper. we have included a future work statement proposing the evaluation of the method's performance for the classification of dermatological diseases using dermoscopy images. We agree that skin lesion detection remains a challenging task, and despite recent advancements, there is still a strong need for an effective AI-based approach in this field. This addition underscores the potential of our method in addressing challenges in other medical imaging domains. we have added a statement suggesting modifications to the proposed method for abdominal organ segmentation, such as the liver and kidneys, from grayscale medical images. We acknowledge that segmentation in such images is often hindered by noise and low contrast, where atlas-based or level-set methods may not be effective. This extension further highlights the adaptability and versatility of the proposed approach. Thank you for improving the completeness of this paper by leaving a comment regarding the shortcomings of this paper.

Modified and supplemented contents – Page 17, 9~15

[31] Goceri, E. Automated Skin Cancer Detection: Where We Are and The Way to The Future. In 2021 44th international conference on telecommunications and signal processing (TSP), (2021) pp. 48-51. https://doi.org/10.1109/TSP52935.2021.9522605.

[32] Göçeri, E. Convolutional neural network based desktop applications to classify dermatological diseases. n 2020 IEEE 4th international conference on image processing, applications and systems (IPAS), (2020) pp. 138-143. https://doi.org/10.1109/IPAS50080.2020.9334956.

[33] Goceri, E., & Karakas, A. A. Comparative evaluations of cnn based networks for skin lesion classification. In 14th International Conference on Computer Graphics, Visualization, Computer Vision and Image Processing (CGVCVIP), Zagreb, Croatia, (2020) pp. 1-6.

[34] Göçeri, E., Ünlü, M. Z., & Dicle, O. A comparative performance evaluation of various approaches for liver segmentation from SPIR images. Turkish Journal of Electrical Engineering and Computer Sciences, 23 (2015) 741-768. https://doi.org/10.3906/elk-1304-36.

[35] Goceri, N., & Goceri, E. A neural network based kidney segmentation from MR images. In 2015 IEEE 14th international conference on machine learning and applications (ICMLA), (2015) pp. 1195-1198. https://doi.org/10.1109/ICMLA.2015.229.

[36] Dura, E., Domingo, J., Göçeri, E., & Martí-Bonmatí, L. A method for liver segmentation in perfusion MR images using probabilistic atlases and viscous reconstruction. Pattern Analysis and Applications, 21 (2018) 1083-1095. https://doi.org/10.1007/s10044-017-0666-z.

[37] Göçeri, E. A comparative evaluation for liver segmentation from spir images and a novel level set method using signed pressure force function. Izmir Institute of Technology (Turkey), (2013).

[38] Goceri, E., Unlu, M. Z., Guzelis, C., & Dicle, O. An automatic level set based liver segmentation from MRI data sets. In 2012 3rd International conference on image processing theory, tools and applications (IPTA), (2012) pp. 192-197. https://doi.org/10.1109/IPTA.2012.6469551.

[39] Goceri, E. Automatic kidney segmentation using Gaussian mixture model on MRI sequences. In Electrical Power Systems and Computers: Selected Papers from the 2011 International Conference on Electric and Electronics (EEIC 2011) in Nanchang, 3 (2011) pp. 23-29. https://doi.org/10.1007/978-3-642-21747-0_4. 

Response to Reviewer 2

We appreciate the reviewer’s insightful comments and constructive feedback, which have helped improve the quality of our manuscript. Below, we provide detailed responses to each comment.

1. Check the typographical error in the last sentence before the conclusion of the introduction. You wrote "om" instead of "on." Then, go through the entire work to rectify typographical errors.

Response

Thank you for identifying the typographical error. The word "om" has been corrected to "on" in the introduction. Additionally, we conducted a thorough review of the manuscript and corrected other typographical errors throughout the document to ensure clarity and precision.

Modified and supplemented contents – Page 4, line 24

2. The part of the introduction containing the literature is scanty and must be elaborated on. There is a lot of work on esophageal cancer using AI, and I think you need to reference or add more of this coupled with the evaluation metric values attained.

Response

In response to the reviewer's comments, we have revised the relevant research section of the introduction to this manuscript as follows. We have expanded the Introduction to include a comprehensive review of related studies using AI to detect esophageal cancer, highlighting the methods, datasets, and evaluation metrics. Wang et al. developed and validated a deep learning algorithm to evaluate the difference in performance of polyp and adenoma detection using colonoscopy, achieving a sensitivity of 94.3% and specificity of 95.9% [18]. Similarly, Xu et al. designed an architecture for real-time classification and detection of gastric polyps through gastroscopy, achieving 100% sensitivity and 95.4% specificity, with excellent performance in detecting small polyps [19]. For esophageal cancer, Goda et al. showed that magnified endoscopy with narrow-band imaging had a sensitivity of 78% and specificity of 95%, comparable to non-magnified high-resolution endoscopy (sensitivity 72%, specificity 92%) and high-frequency endoscopic ultrasound (sensitivity 83%, specificity 89%), and predicted the depth of invasion of superficial esophageal squamous cell carcinoma, reducing the risk of overestimation by 25% compared to other techniques [20]. Nakagawa et al. found that an AI system using a single-shot multi-detector architecture to assess superficial squamous cell carcinoma achieved a sensitivity of 90.1%, specificity of 95.8%, and accuracy of 91%, which was similar to that of an experienced endoscopist, who achieved a sensitivity of 89.8, specificity of 88.3%, and accuracy of 89.6% [21]. Although there have been many CNN-based studies on lesion detection and diagnosis in various organs, medical data for esophageal squamous cell carcinoma is still limited compared to other datasets, which has led to problems such as overfitting and poor performance on new lesion images [22]. Wang et al. reported that Linked Color Imaging had a specificity of 92.4% and sensitivity of 83.7% for esophageal squamous cell carcinoma screening, which was similar to Lugol Chromoendoscopy with a specificity of 87% and sensitivity of 90.7%, and was promising for screening for squamous cell carcinoma and precancerous lesions in the general population with a much shorter procedure time [23].

Modified and supplemented contents – Page 3, line 25, Page 4, line 1~20

3. It would like you to show the confusion matrices that you used to calculate TP, TN, FP, FN, precision, Sensitivity etc. in the tables and in the text.

Response

We have made additions to Tables 2 and 3 of the manuscript based on the reviewer's comments. We have included confusion matrices for both the YOLOv5 and RetinaNet models in the Results section. Thank you for improving the completeness of this paper by leaving a comment regarding the shortcomings of this paper.

Modified and supplemented contents – Page 12, Table 2, Page 14, Table 3

4. If this dataset has been used by any AI algorithm before, a comparative analysis with the authors' results and the existing models' results will help readers and researchers appreciate your efforts.

Response

Thank you for your valuable suggestion for a comparative analysis of our dataset and existing AI models. As per the reviewer's comments, we have carefully reviewed the use of our dataset in previous studies and confirmed that it has not been used by other AI algorithms to date. However, we have included a discussion comparing the performance of our model with similar studies utilizing esophageal endoscopy images. This comparison highlights the strengths of our model and places our work in a broader research context. In addition, as a future direction, we plan to further validate and extend the generalization of our model by applying our methodology to other datasets or imaging modalities, which will allow for broader applicability and comparative studies with other datasets in the future. Thank you again for pointing out this important aspect. It has helped us to improve the manuscript.

5. It would have been good if the Accuracy evaluation metric was also used to assess the models.

Response

Thank you for sharing your insightful comments about using accuracy as an evaluation metric. Accuracy is commonly used in many classification tasks, but for object detection models such as ours, evaluation metrics such as precision, sensitivity, and false positives per image (FPPI) are often prioritized. These metrics are particularly important in detection problems where the goal is not only to classify objects within an image, but to accurately locate them. In detection tasks, datasets often exhibit class imbalance (e.g., fewer positive samples compared to negative samples), which can make accuracy less informative. Metrics such as precision and sensitivity provide a more detailed understanding of a model's detection capabilities, and FPPI helps to assess a model's tendency to generate false positives in a detection setting. However, we appreciate your suggestion, and in future work we will consider incorporating accuracy if it provides more insight into model performance. You raise this important point, which has helped us to reflect on our evaluation approach.

6. You indicated you were proposing an AI algorithm, but I saw the

---

## [Decision Letter · Decision Letter 1]

11 Feb 2025

PONE-D-24-42374R1Early detection of esophageal cancer: evaluating ai algorithms with multi-institutional narrorwband and white-light imaging dataPLOS ONE

Dear Dr. Kim,

Thank you for submitting your manuscript to PLOS ONE. After careful consideration, we feel that it has merit but does not fully meet PLOS ONE’s publication criteria as it currently stands. Therefore, we invite you to submit a revised version of the manuscript that addresses the points raised during the review process. Check comments received from the reviewer 2. 

We look forward to receiving your revised manuscript.

Kind regards,

Hirenkumar Kantilal Mewada

Academic Editor

PLOS ONE

Journal Requirements:

Reviewers' comments:

Reviewer's Responses to Questions

**Comments to the Author**

1. If the authors have adequately addressed your comments raised in a previous round of review and you feel that this manuscript is now acceptable for publication, you may indicate that here to bypass the “Comments to the Author” section, enter your conflict of interest statement in the “Confidential to Editor” section, and submit your "Accept" recommendation.

Reviewer #1: All comments have been addressed

Reviewer #2: (No Response)

2. Is the manuscript technically sound, and do the data support the conclusions?

Reviewer #1: Yes

Reviewer #2: Yes

3. Has the statistical analysis been performed appropriately and rigorously? 

Reviewer #1: Yes

Reviewer #2: I Don't Know

4. Have the authors made all data underlying the findings in their manuscript fully available?

Reviewer #1: Yes

Reviewer #2: No

5. Is the manuscript presented in an intelligible fashion and written in standard English?

Reviewer #1: Yes

Reviewer #2: Yes

6. Review Comments to the Author

Reviewer #1: The proposed approach and the result will be helpful for many researchers.

The paper has been carefully revised by the authors.

After all revisions, the quality of the paper has been improved.

Therefore, the revised version can be accepted for publication.

**Reviewer #2: Comment #1: I only saw the TP, TN, FP, and FN but not the confusion matrices that were used to calculate them. It would be better if those matrices were shown.**

**Comment #2: It would have been good if there were extra novelties highlighted rather than only dataset and hyperparameter tuning.**

7. PLOS authors have the option to publish the peer review history of their article (what does this mean? ). If published, this will include your full peer review and any attached files.

**Do you want your identity to be public for this peer review?** For information about this choice, including consent withdrawal, please see our Privacy Policy .

Reviewer #1: No

Reviewer #2: No

---

## [Author Response · Author response to Decision Letter 1]

24 Feb 2025

Journal Requirements

Response

It is important to write references completely and accurately as required by the journal. We, the authors, fully agree with the journal's requirements, as the references are what support the research in the paper. We have checked and reviewed them thoroughly. We have reviewed the references of the papers and none of them have been retracted. We appreciate your comments regarding the journal to improve the quality of the paper.

Response to Reviewer 1

The proposed approach and the result will be helpful for many researchers. The paper has been carefully revised by the authors. After all revisions, the quality of the paper has been improved. Therefore, the revised version can be accepted for publication.

Response

Thank you for your positive comments on this paper. I think the quality of the paper has improved a lot of thanks to the reviewer's comments. Thank you for your detailed comments on improving the paper.

Response to Reviewer 2

1. I only saw the TP, TN, FP, and FN but not the confusion matrices that were used to calculate them. It would be better if those matrices were shown.

Response

We appreciate the reviewer’s suggestion regarding the confusion matrix. However, in object detection tasks, the concept of True Negative (TN) is not well-defined, as it is in classification tasks. Detection models focus on identifying objects (lesions) within an image, and their performance is typically evaluated using Precision, Sensitivity, FPPI (False Positives Per Image), and mAP (Mean Average Precision) rather than a traditional confusion matrix. In our study, we have already provided a detailed table summarizing TP, FP, and FN values to clearly present the detection performance of our models. Metrics such as precision and sensitivity provide a more detailed understanding of a model's detection capabilities, and FPPI helps to assess a model's tendency to generate false positives in a detection setting. However, we appreciate your suggestion, and in future work we will consider incorporating the confusion matrix if it provides more insight into model performance. You raised this important point, which has helped us to reflect on our evaluation approach.

2. It would have been good if there were extra novelties highlighted rather than only dataset and hyperparameter tuning.

Response

We appreciate the reviewer’s comment regarding the novelty of our study. Beyond dataset expansion and hyperparameter tuning, our work provides significant contributions: (1) multi-institutional dataset validation, improving model generalization and addressing dataset bias; (2) comparative analysis of YOLOv5 and RetinaNet, offering practical insights for clinical applications; (3) integration of both WLI and NBI imaging, enabling a more comprehensive assessment of esophageal cancer detection; (4) detailed analysis of false positives and false negatives, enhancing interpretability and clinical relevance; and (5) proposed future enhancements using synthetic data generation, suggesting broader applications beyond endoscopy. We have clarified these contributions in the revised manuscript. We sincerely appreciate the reviewers' feedback, which helped us to further clarify the novelty and contribution of our work.

---

## [Editor Report · Decision Letter 2]

2 Mar 2025

Early detection of esophageal cancer: evaluating ai algorithms with multi-institutional narrorwband and white-light imaging data

PONE-D-24-42374R2

Dear Dr. Kim,

We’re pleased to inform you that your manuscript has been judged scientifically suitable for publication and will be formally accepted for publication once it meets all outstanding technical requirements.

Kind regards,

Hirenkumar Kantilal Mewada

Academic Editor

PLOS ONE
---

## [Editor Report · Acceptance letter]

PONE-D-24-42374R2

PLOS ONE

Dear Dr. Kim,

I'm pleased to inform you that your manuscript has been deemed suitable for publication in PLOS ONE. Congratulations! Your manuscript is now being handed over to our production team.

Kind regards,

on behalf of

Dr. Hirenkumar Kantilal Mewada

Academic Editor

PLOS ONE